# Research on the path of green technology innovation driven by the Environmental Protection Tax Law: Based on data of heavy polluting enterprises

Wei Tao[1], Jian-ya Zhou[2], Ye-ling Zhao[3]*

1 School of Finance and Business, Zhenjiang College, Zhenjiang, China, 2 School of Management, Jiangsu University, Zhenjiang, China, 3 School of Finance and Economics, Jiangsu University, Zhenjiang, China

* zhaoyel2012@163.com

**Data Availability Statement:** All relevant data are within the manuscript and its Supporting Information files.

## Abstract

Environmental Protection Tax Law (EPTL) is a compulsory environmental regulation measure adopted by China to deal with environmental problems. However, with the advancement of implementation, the EPTL produces a dissimilation effect and damages the realization of the Porter hypothesis effect. The study examines the dissimilation effect of green technology innovation regulated by the EPTL using sample data from heavy pollution firms in China. According to the empirical test results: (1) the coordination between levies and administrations, differential tax rate setting, tax information sharing, definition of the scope of levy and administration, tax declaration counseling, and tax rate level verification produce the dissimilation effect; (2) the Porter hypothesis effect of the EPTL is the most significant in medium-sized enterprises and foreign-funded enterprises. By constructing the research model group of dissimilation effect, this study analyzes the application of environmental regulation in China's social and economic background, thus providing a reference for developing of the green economy.

## 1. Introduction

Due to the rapid economic growth, environmental problems have become a key factor restricting the healthy development of China's economy [1]. The relationship between economic growth and environmental protection is not difficult to reconcile, on the contrary, with the help of necessary environmental policy tools can achieve economic and environmental win-win results [2]. Environmental regulation has become an important policy tool to alleviate environmental problems [3]. China also hopes to achieve the common development of the environment and economy with the help of environmental regulation policy tools. On January 1, 2018, the Chinese EPTL was formally implemented, a major event in the course of China's economic and social development, not only marking the formal establishment of China's environmental protection tax system but also demonstrating the Chinese government's attitude and determination to make economic growth and environmental protection a common

**Funding:** The author(s) received no specific funding for this work.

**Competing interests:** The authors have declared that no competing interests exist.

development objective [4]. The implementation of EPTL in developed countries has a history of more than a hundred years, and rich skills and experience have been accumulated, which has played a positive role in economic growth and environmental protection [5]. However, the normal implementation process of the EPTL in our country has suffered from an abnormal impact and produced the dissimilation effect. In the new economic situation, it is necessary to deeply analyze the dissimilation effect of the EPTL, as well as to propose new corrective and optimization measures to promote the EPTL to play its due value. Thus driving the realization of green technology innovation and effective use of environmental resources [6]. The realization of green technology innovation is of great significance to green economic development and the correction of environmental behavior [7], while the Porter hypothesis effect of EPTL stimulates green technology innovation [8].

The implementation of China's EPTL has been incubated and explored for more than ten years. In May 2007, China's State Council issued the Comprehensive Work Program for Energy Conservation and Emission Reduction, which began to explore the issue of environmental taxation. In June 2007, China's Ministry of Finance (MOF), the General Administration of Taxation (GAT), and the Ministry of Environmental Protection (MEP) set up a Joint Study Group to discuss several issues in the introduction of environmental tax and submitted the draft environmental tax to the State Council at the end of 2014. Subsequently, the Joint Study Group went out for public comment on two more occasions in 2015 and 2016. On December 25, 2016, the Standing Committee of the National People's Congress (NPC) considered and passed the EPTL. On July 21, 2017, China's MOF, GAT, and MEP jointly promulgated the Notice on Comprehensively Preparing for the Implementation of the EPTL. On December 22, 2017, the State Council issued a Notice on the Attribution of Environmental Protection Tax Revenue. On December 25, 2017, the State Council promulgated the Regulations on the Implementation of the EPTL. On December 27, 2017, the MEP issued the Announcement on the Release of Pollution Coefficients and Material Accounting Methods for Calculating Pollutant Emissions. On January 1, 2018, the EPTL came into effect. On January 7, 2018, the MOF issued the Notice on Matters Related to the Suspension of Sewage Charges and Other Administrative Fees. As an important part of the world, China's environmental behavior will be conducive to the improvement of the global environment. China's EPTL is a mandatory environmental governance strategy led by the government [9], and the key is to achieve a win-win situation between the government's environmental concern and corporate financial performance.

As a typical environmental regulation, the EPTL also has the characteristics of the Porter hypothesis effect [10]. Porter (1991) states that under the right market environment, stringent environmental regulations can incentivize firms to implement in-depth innovation and enhance green technological innovation capabilities, partially or completely offsetting the increase in firms' costs under environmental regulations, thus giving firms a competitive advantage in the marketplace, which is the core idea of the Porter hypothesis [11]. Jaffe and Palmer (1997) based on a large number of enterprise environmental regulation case analyses, and further analysis of the Porter hypothesis, argue that if the enterprise's green technology innovation partially offsets the cost increase under environmental regulation, it is called the Weak Porter hypothesis. If enterprises' green technology innovation completely offsets the cost increase under environmental regulation, it is called the Strong Porter hypothesis [12]. It can be seen that the transition from the Weak Porter hypothesis to the Strong Porter hypothesis is the process of gradually realizing the function of environmental regulation. At the same time, more resources can be allocated to sustainable and environmentally friendly projects, to achieve green development [13]. Therefore, environmental regulations will attract more market participants into the green technology market and achieve green innovation [14].

In environmental economic theory, the Porter hypothesis effect is an effective way to test the effectiveness of the implementation of the EPTL. A few studies show that the Porter hypothesis effect of China's EPTL has not yet been realized. Based on the sample data of A-share listed companies in Shanghai and Shenzhen from 2011 to 2019, Yuan (2022) utilized the double-difference method and found that the implementation of the EPTL significantly inhibited the green technological innovation capability of enterprises, especially more profoundly and extensively inhibited the green technological innovation capability of state-owned enterprises [15]. The gap between the implementation of environmental laws and regulations and the original intention of legislation makes China's EPTL seriously ineffective [16], which is not conducive to the mitigation of environmental problems, but also hurts the innovation ability of enterprises. In the face of strict EPTL, enterprises will adopt greenwashing behavior to avoid punishment, which is not only detrimental to the shaping of green technology innovation ability but also brings serious harm to the whole society [17].

However, most studies show that the Porter hypothesis effect of China's EPTL has already appeared, but is in an imperfect state. Wen and Zhong (2020) found that the environmental protection tax reform has a significant Porter hypothesis effect on large and medium-sized enterprises, which significantly promotes the growth of green technology innovation in large and medium-sized enterprises, but has no significant impact on small enterprises. Therefore, it is suggested to implement supporting policies to reduce the adverse impact of environmental protection tax on small and micro enterprises [18]. Cui, Lu, and Wang (2021) took China's A-share listed heavy polluting enterprises from 2014 to 2019 as samples and found that the implementation of the Environmental protection tax significantly increased the investment of China's heavy polluting enterprises and enhanced their green technology innovation, and there was still a large space for the promotion of green technology innovation [19]. Leng and Quan (2021) argued that as an environmental regulation, the EPTL has a natural driving effect on green innovation, but at present, the EPTL only has a driving effect on green innovation in the process of production and operation of enterprises, and it can't yet have a driving effect on the green innovation in the result of production and operation [20]. The focus of environmental protection tax also affects the effect of green technology innovation, but the government paying more attention to environmental issues will hurt the finance of enterprises, and thus endanger the investment in technological innovation [21]. Frequent changes in economic policies have inhibited green technology innovation [22], making it difficult to achieve the goal of environmental protection tax. In the study of environmental regulation policy, some scholars believe that relying solely on the mandatory environmental protection tax is not conducive to the green technology innovation ability [23–25], which makes it difficult to fully realize the Porte hypothesis effect of EPTL.

Some studies show that China's EPTL not only has the Porter hypothesis effect but also the effect of regional differences on the Porter hypothesis can not be ignored. Hu, Chen, and Zhao (2020) tested the panel data of high-polluting manufacturing enterprises in 30 provincial regions in China from 2008 to 2018 and found that environmental protection tax has a significant promoting effect on green technology innovation of high-polluting enterprises, but the Porter hypothesis effect of environmental protection tax lagging one period is more obvious, and pointed out that regional differences are an important factor affecting the effect of Porter hypothesis of environmental protection tax [26]. Also taking Chinese listed companies from 2010 to 2018 as research objects, Kong et al. (2024) believe that in addition to geographical differences, differences in corporate ownership structure will also have an impact on the effect of environmental protection tax [27].

Existing research shows that China's EPTL has shown the Porter hypothesis effect, which to a certain extent has promoted the green technological innovation of enterprises. However,

with the advancement of the process, the promoting effect of green technology innovation is declining. Existing studies have rarely mentioned the alienating effect of environmental protection tax in China, nor have they discussed the internal mechanism in-depth, nor have they conducted a comprehensive review of the promoting effect of environmental protection tax on the technological innovation ability of enterprises. As a compulsory environmental regulation, the environmental regulation behavior and the environmental regulation effect of environmental protection tax have not been fully discussed. Therefore, to implement the EPTL and get rid of the framework of dissimilation effect as soon as possible, it is necessary to deeply analyze the dissimilation mechanism of the environmental protection tax collection and management, and propose improvement plans to improve the quality of the environmental protection tax collection and management, to maximize the green technological innovation and corporate competitive advantage of enterprises.

The innovations of this paper can be summarized as follows: First, the impact of environmental regulations on green technology innovation is explored, revealing the role of government controls on firms' technological development. Second, the green technology innovation driving path based on China's environmental protection tax is explored through empirical tests. Thirdly, the paper analyzes the role of environmental regulation in China's social and economic background under the framework of the dissimilation effect and further verifies the role of the Porter hypothesis in China's environmental background by constructing a research model set of dissimilation effects.

## 2. Research model design of dissimilation effect

### 2.1 Environmental regulation behavior analysis of the EPTL

The EPTL is characterized by both command-and-control environmental regulation and market-incentive environmental regulation, and its environmental regulation is reflected in a series of collection and management activities of local governments, environmental protection agencies, and tax agencies. First, the EPTL itself requires that the tax authorities and the environmental protection authorities cooperate to accomplish the tax collection work, and it requires the establishment of an information-sharing platform between these two departments, clarifying their respective responsibilities and determining the specific content of data sharing. Only the sincere cooperation and coordination between these two key departments can ensure the smooth realization of environmental protection tax collection [28]. Second, the environmental protection department needs to monitor the pollution emissions of enterprises, and then pass the monitoring data to the tax department as the basis for tax calculation. Currently, there are four measurement methods for taxable pollutant emissions, which are the installation of automatic monitoring equipment, entrusting monitoring agencies to implement monitoring, pollutant discharge coefficient and material accounting, and sampling measurement. Environmental protection agencies need to try to achieve scientific monitoring methods and accurate monitoring results to create a good start for the collection of the EPTL [29]. Third, since the level of tax burden directly determines the play of tax leverage, the design of tax rates for environmental protection tax cannot be one-size-fits-all for all industries, and it is necessary to implement different tax rates for different industries, that is, to set up different tax rates for different types of pollutants and pollution levels. The EPTL, based on the principle of statutory taxation, grants localities a certain degree of discretion in the setting of tax rates but sets tax ceilings on different pollutants [30]. Fourth, the construction of the information-sharing platform for environmental protection tax collection and payment has an important supporting role in the smooth implementation of the EPTL, which can not only increase the collaboration between the tax authorities and the environmental protection authorities but

also provide a great convenience for tax-paying enterprises. At the same time, the information sharing platform improves the quality of information disclosure and thus contributes to the fairness of environmental protection tax [31]. Fifth, the scope of environmental protection tax collection theoretically needs to encompass all pollution-emitting industries or enterprises, to ensure the fairness or reasonableness of the environmental protection tax. However, the current tax object of environmental protection tax only includes four categories of air pollutants, water pollutants, solid pollutants, and noise, and adopts the principle of positive enumeration does not enumerate yet not to be levied, As a result, many types of pollutants are omitted, which may lead to the sense of injustice of tax-paying enterprises [32]. Sixth, the complexity of the procedures for filing environmental protection tax returns, leads to difficulties for many enterprises, especially small and medium-sized enterprises, in handling tax returns on their own and calculating tax reductions and other aspects of payment. Tax agencies need to provide counseling to enterprises on their tax payment operations, and gradually improve the business level of taxpayers. Even in large enterprises, taxpayers often find it difficult to cope with the wide variety of tax-related pollutants, complex calculation methods, and specialized tax bases [33]. Seventh, as the collection of environmental protection tax, involves the synergy between several departments, especially between the two key departments of taxation and environmental protection, it is very easy to produce the phenomenon of mutual shirking. Therefore, it is necessary to establish an accountability mechanism to clarify the division of responsibilities of each department and each department's responsibilities, to give full play to the function of collection and management of all kinds of departments [34]. Eighth, under the principle of paying more tax for more emissions and paying less tax for less emissions, the EPTL has set up some preferential policies to encourage enterprises' excessive energy conservation and emission reduction behaviors, but it is difficult to achieve absolute fairness and justice in the implementation process. Therefore, to effectively play the role of preferential policies, tax agencies should ensure that the fairness and impartiality of preferential policies can not go beyond a certain range, otherwise, the preferential policies will not be able to achieve the expected goals [35]. Finally, the theoretical root of the design of environmental protection tax is the theory of Pigouvian tax, that is, the government restrains the emission behavior of enterprises by taxing polluters. If the tax rate or tax amount is properly designed, it will reduce the amount of pollution emitted by enterprises and improve the ecological environment. Specifically, if the amount of environmental protection tax is moderately higher than the cost of sewage treatment for the enterprise, the enterprise will choose to treat the pollution, and vice versa, they will continue to discharge. Of course, the tax level cannot be too high, otherwise it will overwhelm the enterprises. Therefore, in a region, to effectively constrain the emission behavior of most enterprises, it is necessary to set a reasonable level of tax rate [36].

## 2.2 Analysis of the environmental regulation effect of the EPTL

The environmental regulation effect of the EPTL is reflected in either the Weak Porter hypothesis or the Strong Porter hypothesis, but regardless of the state of the hypothesis, the growth of enterprise green technology innovation is an important sign. As long as the enterprise's green technology innovation is improving or enhancing, it means that the environmental regulation of the EPTL is playing a positive or active role [37].

Technological innovation is a traditional economic concept, while green technological innovation is the evolution of technological innovation in the green economic system, responding to the needs of the times of green economic development. Brawn and Wield (1994) first proposed the concept of green technology as a non-polluting technology that utilizes the results of modern science and technology to improve the quality of the environment

[38]. Green technology includes a series of technologies such as environmental protection technology, energy-saving technology, clean production technology, and clean energy technology. The main functions of green technology are to reduce energy consumption, reduce environmental pollution, improve ecology, promote the construction of ecological civilization, and realize the harmonious coexistence of man and nature [39]. Green technology innovation highlights the cultivation, development, and application of green concepts, green products, and green processes, emphasizes the green market as the guide, and promotes the transformation of green technology achievements. Thus it is the combination of green production organization methods, green operation management modes, and green marketing service methods [40].

In current research, green technology innovation is generally divided into three basic elements: green process innovation, green product innovation, and end treatment innovation. Therefore, at the micro level, the environmental regulation effect of the EPTL is reflected in that the collection of environmental protection tax promotes the growth of green process innovation, green product innovation, and end treatment innovation in enterprises [41].

### 2.3 Research model design of dissimilation effect

Under the theoretical framework of Porter hypothesis, based on the analysis of the environmental regulation behavior and environmental regulation effect of the EPTL, and taking into account the influence of the difference in enterprise scale and the difference in enterprise attributes, the research model group on the dissimilation effects of green technology innovation regulated by the EPTL is designed, as shown in the following example. The basic design idea is to first discuss the Porter hypothesis effect of green technology innovation sub-elements, and then discuss the Porter hypothesis effect of green technology innovation as a whole.

$$
\begin{aligned}
Gcra = \beta_{01} + \beta_1 xt + \beta_2 jc + \beta_3 cy + \beta_4 gx + \beta_5 fw + \beta_6 fd + \beta_7 wz + \beta_8 yh + \beta_9 sl + \gamma_1 me + \gamma_2 be \\
+ \gamma_3 pe + \gamma_4 fe + \gamma_5 mi + \gamma_6 we + u_1
\end{aligned} \tag{1}
$$

$$
\begin{aligned}
Gcom = \beta_{02} + \beta_1 xt + \beta_2 jc + \beta_3 cy + \beta_4 gx + \beta_5 fw + \beta_6 fd + \beta_7 wz + \beta_8 yh + \beta_9 sl + \gamma_1 me + \gamma_2 be \\
+ \gamma_3 pe + \gamma_4 fe + \gamma_5 mi + \gamma_6 we + u_2
\end{aligned} \tag{2}
$$

$$
\begin{aligned}
Gend = \beta_{03} + \beta_1 xt + \beta_2 jc + \beta_3 cy + \beta_4 gx + \beta_5 fw + \beta_6 fd + \beta_7 wz + \beta_8 yh + \beta_9 sl + \gamma_1 me + \gamma_2 be \\
+ \gamma_3 pe + \gamma_4 fe + \gamma_5 mi + \gamma_6 we + u_3
\end{aligned} \tag{3}
$$

$$
\begin{aligned}
Gtec = \beta_0 + \beta_1 xt + \beta_2 jc + \beta_3 cy + \beta_4 gx + \beta_5 fw + \beta_6 fd + \beta_7 wz + \beta_8 yh + \beta_9 sl + \gamma_1 me + \gamma_2 be \\
+ \gamma_3 pe + \gamma_4 fe + \gamma_5 mi + \gamma_6 we + u
\end{aligned} \tag{4}
$$

Among them, variable characteristics such as variable name, variable symbol, variable coefficient, and variable connotation are shown in Table 1.

## 3. Research model test of dissimilation effect

### 3.1 Sample survey

Heavy polluting enterprises are a special category of enterprises, which emit large amounts of pollutants in the production process and cause more serious damage to the ecological environment. At the same time, the response to the implementation of the EPTL is also the most direct and sensitive. The main objective of the implementation of China's EPTL is to restrain the discharge of pollutants by heavy polluting enterprises and to promote the realization of the dual

**Table 1. Variable characteristics.**

| Variable name | Variable symbol | Variable coefficient | Variable connotation |
|---|---|---|---|
| **Main independent variable** | | | |
| Coordination between levies and administrations | $xt$ | $\beta_1$ | Coordination between tax levies and environmental authorities in the tax collection process. |
| Emissions monitoring science | $jc$ | $\beta_2$ | The monitoring of pollution emissions from enterprises by the environmental protection authorities is scientific and accurate. |
| Differential tax rate setting | $cy$ | $\beta_3$ | Setting different tax rates or amounts for different industries and different pollutant emissions. |
| Tax information sharing | $gx$ | $\beta_4$ | Information and data on the collection and payment of environmental protection taxes can be shared by both the collector and the payer. |
| Definition of the scope of levy and administration | $fw$ | $\beta_5$ | The subject or scope of the environmental protection tax includes all pollution emissions. |
| Tax declaration counseling | $fd$ | $\beta_6$ | Tax authorities guide enterprises on tax accounting, declaration, reduction, and exemption. |
| Accountability mechanism design | $wz$ | $\beta_7$ | Accountability mechanisms have been put in place and have created good norms for the conduct of levies. |
| Implementation of preferential policies | $yh$ | $\beta_8$ | The preferential policies of environmental protection tax reduction and exemption can be open, just, and fair. |
| Tax rate level verification | $sl$ | $\beta_9$ | The setting of regional tax rates can promote energy conservation and emission reduction for most enterprises. |
| **Controlled variable** | | | |
| Small enterprise | | | Basic variable, do not participate in the model testing. |
| Medium-sized enterprise | $me$ | $\gamma_1$ | Binary variable, the value of 1 indicates that the enterprise is a medium-sized heavy polluting enterprise. |
| Large-sized enterprise | $be$ | $\gamma_2$ | Binary variable, the value of 1 indicates that the enterprise is a large-sized heavy polluting enterprise. |
| State-owned enterprise | | | Basic variable, do not participate in the model testing. |
| Private enterprise | $pe$ | $\gamma_3$ | Binary variable, the value of 1 indicates that the enterprise is a private heavy polluting enterprise. |
| Foreign-funded enterprise | $fe$ | $\gamma_4$ | Binary variable, the value of 1 indicates that the enterprise is a foreign-funded heavy polluting enterprise. |
| Eastern China | | | Basic variable, do not participate in the model testing. |
| Central China | $mi$ | $\gamma_5$ | Binary variable, the value of 1 indicates that the enterprise is located in Central China. |
| Western China | $we$ | $\gamma_6$ | Binary variable, the value of 1 indicates that the enterprise is located in Western China. |
| **Explained variable** | | | |
| Green process innovation | $Gcra$ | | Environmental innovations in process design, production flow, and supply chain of heavily polluting enterprises. |
| Green product innovation | $Gcom$ | | Environmental protection innovation in product transportation, consumption, application, and recycling of heavy polluting enterprises |
| End treatment innovation | $Gend$ | | Environmental innovations in wastewater, gas, sludge treatment, and noise control of heavy polluting enterprises. |
| Green technology innovation | $Gtec$ | | Environmental protection innovation of heavy polluting enterprises in the whole operation process or market competition. |

objectives of economic development and environmental protection. Therefore, this study takes China's heavy polluting enterprises as the object of sample investigation.

According to the Guidelines on Environmental Information Disclosure for Listed Companies published by China's Ministry of Environmental Protection (MEP) on September 14, 2010, heavy polluting industries generally include 16 industries such as metallurgy, iron and steel, cement, building materials, chemical industry, electrolytic aluminum, thermal power, textile, brewing, coal, pharmacy, papermaking, fermentation, tanning, petrifaction, and mining. In actual environmental protection, local governments will also list some special polluting enterprises as heavy polluting industries.

**Table 2. Sample characteristics.**

| Attribute | Type | Sample size | Portion% | Attribute | Type | Sample size | Portion% |
|---|---|---|---|---|---|---|---|
| Sectors Distribution | Metallurgy | 32 | 7.5 | Profit Distribution (¥) | < = 30M | 114 | 26.6 |
| | Iron and steel | 15 | 3.5 | | 30~60M | 98 | 23.0 |
| | Cement | 13 | 3.0 | | 60~90M | 77 | 18.0 |
| | Building materials | 45 | 10.5 | | 90~120M | 62 | 14.5 |
| | Chemical industry | 38 | 8.9 | | 120~150M | 46 | 10.7 |
| | Electrolytic aluminum | 13 | 3.0 | | > = 150M | 31 | 7.2 |
| | Thermal power | 8 | 1.9 | Sales Revenue Distribution (¥) | < = 0.3B | 105 | 24.5 |
| | Textile | 27 | 6.3 | | 0.3~0.6B | 86 | 20.1 |
| | Brewing | 30 | 7.0 | | 0.6~0.9B | 74 | 17.3 |
| | Coal | 19 | 4.4 | | 0.9~1.2B | 71 | 16.6 |
| | Pharmacy | 34 | 8.0 | | 1.2~1.5B | 54 | 12.6 |
| | Papermaking | 14 | 3.3 | | > = 1.5B | 38 | 8.9 |
| | Fermentation | 29 | 6.8 | Output Distribution (¥) | < = 0.3B | 109 | 25.5 |
| | Tanning | 17 | 4.0 | | 0.3~0.6B | 97 | 22.7 |
| | Petrifaction | 16 | 3.7 | | 0.6~0.9B | 81 | 18.9 |
| | Mining | 13 | 3.0 | | 0.9~1.2B | 69 | 16.1 |
| | Other | 65 | 15.2 | | 1.2~1.5B | 47 | 11.0 |
| Region Distribution | East China | 69 | 16.1 | | > = 1.5B | 25 | 5.8 |
| | Dong Nan | 57 | 13.3 | Scale Distribution | Small enterprise | 246 | 57.5 |
| | North China | 64 | 14.9 | | Medium-sized enterprise | 120 | 28.0 |
| | the Northwestern District | 67 | 15.7 | | Large-sized enterprise | 62 | 14.5 |
| | Southwest China | 65 | 15.2 | Attribute Distribution | State-owned enterprises | 95 | 22.2 |
| | Central and Southern China region | 67 | 15.7 | | Private enterprise | 294 | 68.7 |
| | Northeastern China | 39 | 9.1 | | Foreign-funded enterprise | 39 | 9.1 |

Drawing on existing research results, this study first designs the EPTL Environmental Regulation and Green Technology Innovation Questionnaire and then collects data with the help of a 7-point scale system, taking senior executives of heavily polluting companies as survey respondents for the following reasons: First, senior executives can identify and judge the quality, effectiveness, and problems with the implementation of the EPTL from a corporate perspective; Second, senior executives can give a reasonable assessment of the current status and level of green technology innovation capability of enterprises; Third, most senior executives are anxious and hope that the EPTL can benefit the country and the people as soon as possible, so they are willing to fill in the questionnaire according to their intuition and judgment.

This sample survey began on July 5, 2024, and ended on August 15, 2024. With the efforts of all the members of the research group, a total of 428 valid samples were obtained using online questionnaires, and the characteristics of the samples are shown in Table 2. Among them, profit, sales revenue, and output value are all taken from the financial index value of 2023.

The descriptive statistical characteristics of the sample are shown in Table 3.

## 3.2 Research model test

Based on 428 sample data, using Stata15.0 software for the multicollinearity test, found that the research model does not have the problem of multicollinearity. Then, enterprise size type

**Table 3. Descriptive statistical characteristics of samples.**

| Variable name | Maximum value | Minimum value | Mode | Mean value | Variance |
|---|---|---|---|---|---|
| Coordination between levies and administrations ($xt$) | 7 | 1 | 2 | 2.19 | 0.09 |
| Emissions monitoring science ($jc$) | 7 | 1 | 4 | 3.87 | 0.16 |
| Differential tax rate setting ($cy$) | 7 | 1 | 3 | 3.11 | 0.12 |
| Tax information sharing ($gx$) | 7 | 1 | 2 | 2.29 | 0.10 |
| Definition of the scope of levy and administration ($fw$) | 7 | 1 | 3 | 2.98 | 0.13 |
| Tax declaration counseling ($fd$) | 7 | 1 | 2 | 1.99 | 0.07 |
| Accountability mechanism design ($wz$) | 7 | 1 | 3 | 3.08 | 0.12 |
| Implementation of preferential policies ($yh$) | 7 | 1 | 4 | 3.91 | 0.15 |
| Tax rate level verification ($sl$) | 7 | 1 | 2 | 2.22 | 0.08 |
| Green process innovation ($Gcra$) | 7 | 1 | 3 | 2.88 | 0.13 |
| Green product innovation ($Gcom$) | 7 | 1 | 3 | 3.03 | 0.16 |
| End treatment innovation ($Gend$) | 7 | 1 | 3 | 3.27 | 0.10 |
| Green technology innovation ($Gtec$) | 7 | 1 | 3 | 2.91 | 0.12 |

was selected as the control variable to conduct the model test, and the test results are shown in Table 4.

Based on the initial test of the research model, the enterprise attribute type and area attribute type are continued to be added as control variables, and the robustness test of the research model is carried out, the test results are shown in Table 5. A comparison of Tables 5 with 4 reveals that the model test is robust.

## 3.3 Discussion

The study empirically tests the effects of the Porter hypothesis of the EPTL by constructing a research model set of the dissimilation effects of green technological innovations regulated by the EPTL and utilizing sample data from heavy polluting enterprises in China. According to the results of the model test, it can be seen that with the implementation of China's EPTL, the promotion effect of environmental regulation on green technological innovation has been alienated, and the effect of Porter hypothesis has failed to be fully realized, and the realization of the green technological innovation ability of heavy polluting enterprises has been impeded by the existing regulatory behavior.

The environmental regulation effect of the EPTL has been proved [42], and the effective implementation of the tax law can urge enterprises to take positive measures to deal with environmental punishment, to increase their investment in green technology innovation [43]. However, the micro-mechanism of the Porter hypothesis effect of the EPTL in the specific implementation process has not yet been discussed in depth, nor has there been any analysis of the regulatory behavior of the EPTL, nor has there been any exploration of whether the existing regulatory behavior is conducive to the green technological innovation capability of enterprises. Previous studies believe that compulsory taxation can improve the ESG performance of

**Table 4. Test results of the research model.**

| | Green process innovation (*Gcra*) | Green product innovation (*Gcom*) | End treatment innovation (*Gend*) | Green technology innovation (*Gtec*) |
|---|---|---|---|---|
| Coordination between levies and administrations (*xt*) | 0.05(0.0632) | 0.03(0.0702) | 0.12**(0.0072) | 0.07(0.0611) |
| Emissions monitoring science (*jc*) | 0.12**(0.0063) | 0.13***(0.0005) | 0.15***(0.0008) | 0.14***(0.0003) |
| Differential tax rate setting (*cy*) | 0.09*(0.0315) | 0.07(0.0858) | 0.11**(0.0076) | 0.10*(0.0332) |
| Tax information sharing (*gx*) | 0.02(0.1121) | 0.01(0.0871) | 0.03(0.0772) | 0.00(0.0518) |
| Definition of the scope of levy and administration (*fw*) | 0.08*(0.0405) | 0.10**(0.0059) | 0.07(0.0606) | 0.09*(0.0323) |
| Tax declaration counseling (*fd*) | 0.07(0.0922) | 0.05(0.0688) | 0.09*(0.0303) | 0.03(0.0627) |
| Accountability mechanism design (*wz*) | 0.11**(0.0059) | 0.09*(0.0401) | 0.13***(0.0005) | 0.10**(0.0027) |
| Implementation of preferential policies (*yh*) | 0.13***(0.0006) | 0.08*(0.0328) | 0.15***(0.0008) | 0.12***(0.0006) |
| Tax rate level verification (*sl*) | 0.02(0.1031) | 0.03(0.0962) | 0.10*(0.0356) | 0.06(0.0708) |
| Medium-sized enterprise (*me*) | 0.12**(0.0059) | 0.07(0.0602) | 0.14***(0.0002) | 0.13***(0.0004) |
| Large-sized enterprise (*be*) | 0.10*(0.0202) | 0.05(0.0662) | 0.11***(0.0007) | 0.09*(0.0215) |
| $R^2$ | 0.69 | 0.70 | 0.62 | 0.66 |
| $\Delta R^2$ | 0.01 | 0.02 | 0.02 | 0.01 |
| adjusted $R^2$ | 0.70 | 0.72 | 0.64 | 0.67 |
| adjusted *F* Value | 108.36 | 88.19 | 79.26 | 138.76 |
| *P* Value | **(0.0033) | ***(0.0004) | **(0.0052) | ***(0.0005) |

Notes: *$P<0.05$

**$P<0.01$

***$P<0.001$; N = 428; The values in parentheses are P value.

enterprises and drive the realization of enterprise innovation [44]. However, some studies believe that too stringent environmental regulations will hinder the development of green productivity of enterprises [45, 46], reduce the financial performance of enterprises, and squeeze the R&D resources of green technologies [47]. With the maturity of the EPTL system, more and more non-punitive policy tools have begun to play a role, complementing the mandatory tax law, and mobilizing the enthusiasm of enterprises to implement green technology innovation [48, 49]. However, the regulatory behavior of the EPTL still plays a major role. Through the study of the existing regulatory behavior, the weak points in the enforcement system of the tax law can be found to better activate the Porter hypothesis effect of the EPTL and promote the development of enterprises' green technology innovation ability.

According to the empirical results of the research model group on the dissimilation effect of green technology innovation regulated by the EPTL, it can be found that some regulatory behaviors are difficult to promote the growth of enterprises' green technology innovation ability. Specifically, the coordination between levies and administrations is only conducive to the realization of end treatment innovation, indicating that there are coordination problems between tax departments and environmental protection departments in the process of tax collection during the implementation of EPTL. For enterprises, meeting the different requirements of the tax department and the environmental protection department in the tax

**Table 5. Test results of the research model (robustness test).**

| | Green process innovation (*Gcra*) | Green product innovation (*Gcom*) | End treatment innovation (*Gend*) | Green technology innovation (*Gtec*) |
|---|---|---|---|---|
| Coordination between levies and administrations (*xt*) | 0.07(0.1032) | 0.05(0.1164) | 0.12**(0.0062) | 0.08(0.0645) |
| Emissions monitoring science (*jc*) | 0.12**(0.0084) | 0.13***(0.0007) | 0.12**(0.0067) | 0.11**(0.0052) |
| Differential tax rate setting (*cy*) | 0.10*(0.0427) | 0.03(0.0976) | 0.13**(0.0084) | 0.10*(0.0382) |
| Tax information sharing (*gx*) | 0.05(0.1028) | 0.01(0.0981) | 0.05(0.0655) | 0.03(0.0852) |
| Definition of the scope of levy and administration (*fw*) | 0.11*(0.0452) | 0.13**(0.0086) | 0.07(0.0583) | 0.13*(0.0355) |
| Tax declaration counseling (*fd*) | 0.07(0.1076) | 0.05(0.0927) | 0.11*(0.0337) | 0.05(0.0876) |
| Accountability mechanism design (*wz*) | 0.11**(0.0052) | 0.12*(0.0257) | 0.13**(0.0076) | 0.11*(0.0411) |
| Implementation of preferential policies (*yh*) | 0.10**(0.0064) | 0.11*(0.0322) | 0.12**(0.0083) | 0.14***(0.0007) |
| Tax rate level verification (*sl*) | 0.02(0.0623) | 0.05(0.0877) | 0.10*(0.0201) | 0.06(0.0768) |
| Medium-sized enterprise (*me*) | 0.15***(0.0004) | 0.07(0.0966) | 0.13**(0.0064) | 0.15***(0.0006) |
| Large-sized enterprise (*be*) | 0.14**(0.0065) | 0.02(0.0662) | 0.12**(0.0072) | 0.13*(0.0352) |
| Private enterprise (*pe*) | -0.13***(0.0005) | -0.11**(0.0056) | -0.12**(0.0084) | -0.14***(0.0006) |
| Foreign-funded enterprise (*fe*) | 0.10*(0.0266) | 0.11*(0.0314) | 0.12**(0.0056) | 0.16***(0.0004) |
| Central China (*mi*) | -0.08(0.0866) | -0.10*(0.0288) | -0.02(0.0705) | -0.12**(0.0064) |
| Western China (*we*) | -0.11**(0.0047) | -0.02(0.1035) | -0.14***(0.0005) | -0.02(0.0688) |
| $R^2$ | 0.72 | 0.68 | 0.66 | 0.67 |
| $\Delta R^2$ | 0.05 | 0.02 | 0.02 | 0.04 |
| adjusted $R^2$ | 0.77 | 0.70 | 0.68 | 0.71 |
| adjusted $F$ Value | 123.29 | 92.78 | 100.26 | 87.19 |
| $P$ Value | **(0.0067) | **(0.0026) | ***(0.0008) | ***(0.0005) |

Notes: *$P < 0.05$

**$P < 0.01$

***$P < 0.001$; N = 428; The values in parentheses are P value.

collection process increases the cost of enterprises virtually. Differential tax rate setting is not conducive to the realization of green product innovation, indicating that the existing tax rate level is relatively reasonable, but it is unable to implement a more effective tax rate for product differentiation. Tax information sharing behavior is in a failed state, indicating that it is difficult for enterprises and governments to realize information sharing at the level of environmental protection tax information, thus providing space for the birth of opportunistic behaviors. Definition of the scope of levy and administration behavior is not conducive to the realization of end treatment innovation, which means that the existing environmental protection tax pays insufficient attention to the treatment of wastewater, gas, and slag after production and noise control, and more supervision content is placed on the pre-production and mid-production process.

Tax declaration counseling only affects the end treatment innovation, indicating that the quality of the existing staff is a big problem and can not meet the requirements of

environmental regulations on personnel ability. Tax rate level verification only affects the end treatment innovation, which means that the current tax rate of environmental protection tax makes it difficult to stimulate the green technology innovation behavior of most enterprises, the region lacks certain flexibility in the tax rate setting level, and the regional differences are not well reflected in the EPTL. The robustness test results after adding control variables can be found that large and medium-sized enterprises lack green product innovation, and the reason behind this is that the cost of product adjustment for large and medium-sized enterprises exceeds that of small enterprises, technological changes require more resource input, and the implementation of environmental protection tax has squeezed the resources of enterprises, thus making it difficult for green product innovation to reach the expected level. From the perspective of enterprise attributes, the EPTL of foreign-funded enterprises has the most significant regulatory effect, indicating that foreign-funded enterprises have more experience in environmental tax than domestic enterprises, and can quickly adjust corporate behavior according to foreign experience. However, private enterprises do not adapt to the implementation of the EPTL, and the profit-oriented corporate business strategy makes them more inclined to take cunning actions to deal with the environmental tax. From regional perspective, environmental regulation in the central and western regions is not conducive to the realization of enterprises' green technology innovation ability. The reason is that compared with the eastern region, the development of enterprises in the central and western regions depends more on natural resources, and the regulatory behavior has insufficient restraint on regional enterprise behavior, which makes it difficult to stimulate the Porter hypothesis effect.

To achieve environmental protection by using enterprise green technology innovation is the goal of implementing the EPTL, and also the key to the effect of environmental regulation. However, from the empirical test results, the implementation of EPTL at the present stage has produced dissimilation, which seriously affects the realization of green technology innovation ability of enterprises. Although some regulatory behaviors are still playing a role, they have brought serious threats to the transformation and upgrading of enterprise technology structure as a whole, which is not conducive to the sustainable development ability of enterprises, but also makes the EPTL lose its due role. Therefore, to better guide the green technology innovation ability of enterprises and drive the green economy, the regulatory behavior of EPTL needs to be optimized to realize the application of Porter hypothesis effect in the management context of China.

## 4. Conclusions and policy implications

### 4.1 Conclusions

Through the comprehensive judgment of the test results, the following conclusions can be drawn. First, the environmental regulatory acts of the EPTL, such as the coordination of collection and management departments, the sharing of tax collection information, the counseling on tax declaration, and the approval of tax rate levels, have not produced a substantial effect on the promotion of enterprises' green technological innovation, while the rest of the regulatory acts have facilitated the growth of the enterprises' green technological innovation capacity to varying degrees. Second, the Porter hypothesis effect of the EPTL is most pronounced in medium-sized enterprises, weakest in large-sized enterprises, and second weakest in small enterprises. Third, the Porter hypothesis effect of the EPTL is the most obvious in foreign-funded enterprises, followed by state-owned enterprises, and the weakest in private enterprises.

In conclusion, with the implementation of the EPTL, regulatory behavior difficult to ensure the effectiveness of the EPTL. The main performance is Porter hypothesis effect has been

significantly weakened, the driving force for enterprises' green technology innovation is decreased, and most regulatory behaviors hinder the occurrence of enterprises' green technology innovation. Therefore, this unfavorable situation should be reversed as soon as possible, the Porter hypothesis effect should be restored, and the environmental regulatory function of the EPTL should be fully realized, to vigorously enhance the green technological innovation capability of enterprises.

By constructing the dissimilarity effect research model group to empirically test the green technology innovation driving path of China's environmental protection tax, this study expands the application of the Porter hypothesis effect in China's management context, further reveals the micro-mechanism of the environmental regulatory effect of the EPTL, and provides theoretical references and practical bases for the realization of the co-development of the economy and the environment for the rest of the world countries and regions.

## 4.2 Policy implications

According to the test results, combined with the investigation on the implementation of China's EPTL, it can be seen that to more effectively drive the growth of green technological innovation in enterprises, the reform and improvement of the EPTL should be focused on promoting in the following directions. First, strengthen the contact and communication between the tax department and the environmental protection department at the senior management. According to the empirical test results, the coordination between levies and administrations has not achieved the drive of green process innovation, green product innovation, and green technology innovation. Therefore, it is necessary to break the business gap between the tax department and the environmental protection department, improve the coordination in the operation process, and ensure the effect of the implementation of the EPTL. Second, local government departments need to vigorously promote the construction of information sharing platforms for environmental protection tax. According to the empirical test results, the regulatory behavior of tax information sharing does not play a driving role in the green technology innovation ability of enterprises. Local governments need to personally guide environmental protection agencies, tax agencies, and tax-paying enterprises to participate together, to vigorously promote the process of information sharing platform construction. Third, the tax authorities need to increase the business training of enterprise taxpayers. According to the empirical test results, the regulation behavior of tax declaration counseling has a serious alienation effect, which only plays a role in the end treatment innovation. Therefore, the tax authorities need to send experienced tax personnel to participate in the training work and change the training method to open the training situation as soon as possible. Finally, flexibly adjust the tax rate level of regional environmental protection tax. According to the empirical test results, the regulatory behavior of tax rate level verification is conducive to end-treatment innovation, but not conducive to green process innovation, green product innovation, and green technology innovation. Therefore, each region should establish the most effective tax rate according to their economy, society, and environmental protection, to promote the dual development of environmental protection and economic growth.

## 5. Limitations and future prospects

Based on econometrics, Porter hypothesis, and environmental regulation, this paper studies the driving path of green technology innovation in the regulation of the EPTL. Based on the data of China's heavy polluting enterprises, this paper analyzes the deficiencies of government environmental regulation in the process of enterprise green technology innovation and development. However, limited to the researcher's knowledge framework, the research has

shortcomings. (1) The target of the survey is the heavy polluting enterprises in China, so it is not enough to pay attention to the light industry and even other industries. In the face of China's dual-carbon development goal, it needs the joint efforts of the whole society, and it cannot only focus on the technological development of enterprises in a certain field. (2) The research area is relatively narrow, and only the environmental protection tax system implemented in China is studied, without comparative analysis with the relevant policies of other countries in the world, and there are limitations in the research conclusions.

To better study the driving effect of the government environmental system on enterprises' green technology innovation capability, the follow-up research will be carried out from the following directions. According to the limitations of the research object, the characteristics of technological innovation of enterprises in more fields can be included under the framework of environmental regulation analysis, to expand the scope of research on environmental regulation. At the same time, it is also possible to study the driving effect of different environmental regulations on enterprises' green technological innovation with the help of the comparative study of the differences in environmental policies in different regions, to further optimize the research framework of environmental regulation.

## Supporting information

**S1 Data.**
(DOC)

## Author Contributions

**Conceptualization:** Wei Tao.

**Data curation:** Wei Tao.

**Formal analysis:** Ye-ling Zhao.

**Investigation:** Wei Tao, Jian-ya Zhou.

**Methodology:** Wei Tao, Jian-ya Zhou, Ye-ling Zhao.

**Project administration:** Ye-ling Zhao.

**Software:** Jian-ya Zhou.

**Supervision:** Jian-ya Zhou, Ye-ling Zhao.

**Validation:** Wei Tao, Ye-ling Zhao.

**Writing – original draft:** Wei Tao.

**Writing – review & editing:** Jian-ya Zhou, Ye-ling Zhao.

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
