## [Decision Letter · Decision Letter 0]

4 Jul 2024

PONE-D-24-20667Research on the green technology innovation-driven path regulated by the Environmental Protection Tax Law in the post-epidemic eraPLOS ONE

Dear Dr. Zhao,

Thank you for submitting your manuscript to PLOS ONE. After careful consideration, we feel that it has merit but does not fully meet PLOS ONE’s publication criteria as it currently stands. Therefore, we invite you to submit a revised version of the manuscript that addresses the points raised during the review process.

We look forward to receiving your revised manuscript.

Kind regards,

Bingnan Guo

Academic Editor

PLOS ONE

Journal Requirements:

Additional Editor Comments:

1. the Introduction and Literature Review sections do not do a good job of highlighting the marginal contributions of this paper and need to be reworked.

2. the model has a lot of independent variables and is very prone to the problem of multicollinearity, and the article lacks relevant tests.

3. only the correlation coefficient values are shown in Table 3, usually, the t-value statistic or standard deviation of the coefficient values are also needed to be shown, which is recommended to be added.Other forms have the same problem.

4. the recommendations of the article are a bit long and it is recommended to analyse them in conjunction with the conclusions.

5. the reviewer's recommended literature is optional, if it is not relevant, there is no need to force the citation of irrelevant literature.

Reviewers' comments:

Reviewer's Responses to Questions

**Comments to the Author**

1. Is the manuscript technically sound, and do the data support the conclusions?

Reviewer #1: Yes

Reviewer #2: Yes

2. Has the statistical analysis been performed appropriately and rigorously? 

Reviewer #1: Yes

Reviewer #2: No

3. Have the authors made all data underlying the findings in their manuscript fully available?

Reviewer #1: No

Reviewer #2: Yes

4. Is the manuscript presented in an intelligible fashion and written in standard English?

Reviewer #1: No

Reviewer #2: No

5. Review Comments to the Author

Reviewer #1: This paper is interesting. My comments are as below

1. Introduction of the paper is weak and does not have flow,

2. Abstract needs to rewrite

3. Discussion part is required to rewrite, author is expected to discuss the inferences of analysis

4. Literature may be updated by including following studies

https://doi.org/10.1177/00194662211062428

https://doi.org/10.1002/sd.2797

https://doi.org/10.1016/j.jclepro.2023.139355

https://doi.org/10.1177/0958305X231183921

https://doi.org/10.1108/IJESM-02-2019-0007

https://doi.org/10.1177/2278682117713577

https://doi.org/10.1108/WJSTSD-01-2018-0007

https://doi.org/10.1007/s11356-023-25317-w

https://doi.org/10.1177/0974929217744462

https://doi.org/10.1080/00927678.2019.1639003

https://doi.org/10.1177/2278682116649844

Language is required through revision.

Reviewer #2: For this article, I have the following suggestions:

Title

The title "Research on the Path of Green Technology Innovation Driven by the Post-pandemic Era under the Regulation of the Environmental Protection Tax Law" is too broad. The author is advised to reflect the research object and specific content in the title, such as "Research on the Path of Green Technology Innovation Driven by the Environmental Protection Tax Law based on Data of Heavily Polluting Enterprises", to better reflect the actual research content of the article.

Abstract

The abstract still does not accurately summarize the content of the article. The author is advised to re-edit the abstract, highlighting the innovation points of the research, the research methods, and the main findings, while controlling it within 200 words.

Theoretical Basis

The analysis of environmental regulation behavior and environmental regulation effect is relatively comprehensive, but too general. The author is advised to simplify and focus the theoretical basis in combination with the research background and data objects of the article. At the same time, the introduction to environmental tax law can be properly simplified.

Research Design

(1) In terms of sample selection, the author's approach of having executives of heavily polluting enterprises fill out questionnaires is reasonable, but the description of sample characteristics is too simple. The author is advised to supplement more descriptive statistical data of the samples.

(2) In terms of variable setting, the control variables only consider the size and attributes of the enterprise. The author is advised to further expand the scope of control variables, such as regional differences and industry differences, to improve the accuracy and credibility of the research.

(3) In terms of model construction, the author considers the two levels of environmental regulation behavior and effect, and the research ideas are clear. However, the details of specific variable setting and model formula are not sufficiently elaborated. The author is advised to further explain and improve this part.

Empirical Results

(1) The results reporting is not rigorous enough. The author directly lists the coefficient values of the model regression results, but lacks necessary statistical explanations of these values, such as the significance level of the coefficients.

(2) The results analysis lacks depth. The author only makes a simple description of the regression results, without in-depth discussion of the mechanism of action of different variables and their economic significance.

(3) The setting of the robustness test is relatively reasonable, but the interpretation of the results is also insufficient.

Conclusions and Policy Recommendations

(1) The conclusion part summarizes the empirical findings comprehensively, but the author is advised to grade the importance of each conclusion point, and better highlight the main findings.

(2) The policy recommendations part puts forward some reasonable insights, but overall the policy recommendations are still thin, lacking necessary details and quantitative analysis support.

(3) Although the innovation points are summarized, the discussion is not concise enough. The author is advised to refine the innovation points in 1-2 core sentences.

Language and Format

(1) There are some problems with the format of data expression and formula numbering, and the author is advised to unify the format.

(2) The normative of the cited references needs to be further improved.

6. PLOS authors have the option to publish the peer review history of their article (what does this mean?). If published, this will include your full peer review and any attached files.

Reviewer #1: No

Reviewer #2: No

---

## [Author Response · Author response to Decision Letter 0]

16 Jul 2024

Additional Editor Comments:

1. the Introduction and Literature Review sections do not do a good job of highlighting the marginal contributions of this paper and need to be reworked.

2. the model has a lot of independent variables and is very prone to the problem of multicollinearity, and the article lacks relevant tests.

3. only the correlation coefficient values are shown in Table 3, usually, the t-value statistic or standard deviation of the coefficient values are also needed to be shown, which is recommended to be added.Other forms have the same problem.

4. the recommendations of the article are a bit long and it is recommended to analyse them in conjunction with the conclusions.

5. the reviewer's recommended literature is optional, if it is not relevant, there is no need to force the citation of irrelevant literature.

Amendments

1.The author has revised the introduction and literature review.

2.Before the study of model testing, multicollinearity tests were carried out, and both were described in the paper. For fear of too much space, the correlation table was not placed in the paper.

3. After the relative value, the p-value is indicated, which also indicates significance and has the same effect as the T-value. 

4.The author has rewritten the conclusions and countermeasures.

5.The author makes reference to some relevant literatures.

Reviewer #1: This paper is interesting. My comments are as below

1. Introduction of the paper is weak and does not have flow

2.Abstract needs to rewrite

3.Discussion part is required to rewrite, author is expected to discuss the inferences of analysis

4. Literature may be updated by including following studieshttps://doi.org/10.1177/00194662211062428

https://doi.org/10.1002/sd.2797

https://doi.org/10.1016/j.jclepro.2023.139355

https://doi.org/10.1177/0958305X231183921

https://doi.org/10.1108/IJESM-02-2019-0007

https://doi.org/10.1177/2278682117713577

https://doi.org/10.1108/WJSTSD-01-2018-0007

https://doi.org/10.1007/s11356-023-25317-w

https://doi.org/10.1177/0974929217744462

https://doi.org/10.1080/00927678.2019.1639003

https://doi.org/10.1177/2278682116649844

Language is required through revision.

Amendments 

1. The author has revised the introduction to achieve higher quality.

2.The author has rewritten the abstract.

3.The author reorganizes the discussion part and analyzes the economic significance behind it.

4.The author has adjusted the references and partially cited the references provided.

5.The author has adjusted the language to meet the requirements of language quality.

Reviewer #2: For this article, I have the following suggestions:

Title

The title "Research on the Path of Green Technology Innovation Driven by the Post-pandemic Era under the Regulation of the Environmental Protection Tax Law" is too broad. The author is advised to reflect the research object and specific content in the title, such as "Research on the Path of Green Technology Innovation Driven by the Environmental Protection Tax Law based on Data of Heavily Polluting Enterprises", to better reflect the actual research content of the article.

Abstract

The abstract still does not accurately summarize the content of the article. The author is advised to re-edit the abstract, highlighting the innovation points of the research, the research methods, and the main findings, while controlling it within 200 words.

Theoretical Basis

The analysis of environmental regulation behavior and environmental regulation effect is relatively comprehensive, but too general. The author is advised to simplify and focus the theoretical basis in combination with the research background and data objects of the article. At the same time, the introduction to environmental tax law can be properly simplified.

Research Design

(1) In terms of sample selection, the author's approach of having executives of heavily polluting enterprises fill out questionnaires is reasonable, but the description of sample characteristics is too simple. The author is advised to supplement more descriptive statistical data of the samples.

(2) In terms of variable setting, the control variables only consider the size and attributes of the enterprise. The author is advised to further expand the scope of control variables, such as regional differences and industry differences, to improve the accuracy and credibility of the research.

(3) In terms of model construction, the author considers the two levels of environmental regulation behavior and effect, and the research ideas are clear. However, the details of specific variable setting and model formula are not sufficiently elaborated. The author is advised to further explain and improve this part.

Empirical Results

(1) The results reporting is not rigorous enough. The author directly lists the coefficient values of the model regression results, but lacks necessary statistical explanations of these values, such as the significance level of the coefficients.

(2) The results analysis lacks depth. The author only makes a simple description of the regression results, without in-depth discussion of the mechanism of action of different variables and their economic significance.

(3) The setting of the robustness test is relatively reasonable, but the interpretation of the results is also insufficient.

Conclusions and Policy Recommendations

(1) The conclusion part summarizes the empirical findings comprehensively, but the author is advised to grade the importance of each conclusion point, and better highlight the main findings.

(2) The policy recommendations part puts forward some reasonable insights, but overall the policy recommendations are still thin, lacking necessary details and quantitative analysis support.

(3) Although the innovation points are summarized, the discussion is not concise enough. The author is advised to refine the innovation points in 1-2 core sentences.

Language and Format

(1) There are some problems with the format of data expression and formula numbering, and the author is advised to unify the format.

(2) The normative of the cited references needs to be further improved.

 Amendments 

1.The author changed the title to reflect the research content.

2. The author has rewritten the abstract.

3.In the introduction of Section 1, the relevant content is simplified.

4.The data of sample descriptive statistical analysis were supplemented.

5. In order to further improve the accuracy and credibility of the study, regional difference was added as a control variable in the robustness test, and the research model was re-tested.

6.The details of the model formula and the attributes of the variables have been explained in detail in Table 1 Variable characteristics.

7.The significance levels of the coefficients have been described and illustrated using p-values. If you use words to elaborate, it will make the paper appear bloated.

8.The author completed the discussion of the result analysis by supplementing the discussion part.

9. The results are explained in the discussion section.

10. The authors reframe the conclusion, highlighting the main findings.

11.The author made some changes to the countermeasure part and rewrote it according to the opinions of other reviewers.

12. The author re-summarizes the innovative points to make it more concise.

13.The authors unify the format of data expression and formula number.

14.According to the requirements of the journal, the author has perfected the standard of the cited references.

---

## [Editor Report · Decision Letter 1]

19 Jul 2024

Research on the path of green technology innovation driven by the Environmental Protection Tax Law: Based on data of heavy polluting enterprises

PONE-D-24-20667R1

Dear Dr. Zhao,

We’re pleased to inform you that your manuscript has been judged scientifically suitable for publication and will be formally accepted for publication once it meets all outstanding technical requirements.

Kind regards,

Bingnan Guo

Academic Editor

PLOS ONE
---

## [Editor Report · Acceptance letter]

22 Jul 2024

PONE-D-24-20667R1 

PLOS ONE

Dear Dr. Zhao, 

I'm pleased to inform you that your manuscript has been deemed suitable for publication in PLOS ONE. Congratulations! Your manuscript is now being handed over to our production team.

Kind regards, 

on behalf of

Professor Bingnan Guo 

Academic Editor

PLOS ONE